# Endometriosis and Infertility: A Long-Life Approach to Preserve Reproductive Integrity

**DOI:** 10.3390/ijerph19106162

**Published:** 2022-05-19

**Authors:** Maria Elisabetta Coccia, Luca Nardone, Francesca Rizzello

**Affiliations:** 1Department of Biomedical, Experimental and Clinical Sciences “Mario Serio”, University of Florence, 50134 Florence, Italy; nardonel@aou-careggi.toscana.it; 2Assisted Reproductive Technology Centre, Careggi University Hospital, 50134 Florence, Italy; francesca.rizzello@gmail.com

**Keywords:** endometriosis, infertility, ART, surgery, endometrioma

## Abstract

Laparoscopic surgery was originally considered the gold standard in the treatment of endometriosis-related infertility. Assisted reproductive technology (ART) was indicated as second-line treatment or in the case of male factor. The combined approach of surgery followed by ART proved to offer higher chances of pregnancy in infertile women with endometriosis. However, it was highlighted how pelvic surgery for endometriosis, especially in cases of ovarian endometriomas, could cause iatrogenic damage due to ovarian reserve loss, adhesion formation (scarring), and ischemic damage. Furthermore, in the last few years, the trend to delay the first childbirth, recent technological advances in ultrasound diagnosis, and technological progress in clinical and laboratory aspects of ART have certainly influenced the approach to infertility and endometriosis with, ART assuming a more relevant role. Management of endometriosis should take into account that the disease is chronic and involves the reproductive system. Consequently, treatment and counselling should aim to preserve the chances of pregnancy for the patient, even if it is not associated with infertility. This review will analyse the evolution of the management of infertility associated with endometriosis and propose an algorithm for treatment decision-making based on the most recent acquisitions.

## 1. Introduction

Endometriosis is defined as the presence of endometrial tissue in sites other than the uterine cavity. It may involve ovaries, fallopian tubes, and the pelvis and has been associated with chronic pain, dysmenorrhoea, dyspareunia, and infertility. The association between endometriosis and infertility is well supported throughout the literature, but a precise cause-and-effect relationship is still controversial.

The estimated overall prevalence of endometriosis is 6–10% of the general female population, peaking between 25–35 years [1], and an annual incidence of 0.1% among women aged 15–49 years [2]. Almost 30–80% of patients with chronic pelvic pain and dysmenorrhoea are diagnosed with endometriosis. In infertile women, the prevalence of endometriosis varies from 20 to 50%, and 35 to 50% of women with endometriosis are infertile [3].

## 2. Classification and Histopathology

Endometriotic lesions have various aspects and degrees of the extent of their severity. Since 1979, classifications have been proposed for the purpose of describing the stage of pathology. However, these classification systems did not provide an adequate prognosis on the reproductive aspect (American Fertility Society (AFS), American Society for Reproductive Medicine (ASRM)). According to Nisolle et al., three types of endometriotic lesions must be considered with different morphology and pathogenesis: peritoneal endometriosis, ovarian endometriosis, and deep infiltrating endometriosis [4].

The Enzian classification was proposed in order to provide a morphologically descriptive classification of deeply infiltrating lesions as a complement to the revised ASRM [5]. However, it has not been widely used in clinical practice.

The management of women with infertility must certainly take into account the different forms of endometriosis. Adamson et al. (2010) proposed the Endometriosis Fertility Index (EFI) score for predicting the chances of spontaneous pregnancy after surgery [6]. This score is based on historical and surgical factors. A point score of 0–3 corresponds to a 10% probability of spontaneous pregnancy after three years. The highest score of 9–10 points is associated with 75% probability. Similar results were found in External validation studies of the EFI [7].

More recently, Ferrier et al. (2020), in a cost-effectiveness study based on the EFI score, concluded that In vitro fertilization—Intracytoplasmic Sperm Injection (IVF-ICSI) after surgery showed effectiveness but with a significant increase in costs for the healthcare system [8].

From the World Endometriosis Consensus held in 2014, it was established that the most adequate classification system probably brings together several classification systems, including the rASRM and the Enzian classifications, along with the EFI [9].

## 3. Endometriosis and Infertility

The relation between endometriosis and infertility is clinically recognized and well supported throughout the evidence, although a definitive cause-effect connection is debatable. Endometriosis-associated infertility is considered a multifactorial problem related to altered immunity and genetics that affects not only the fallopian tubes and embryo transport but also the endometrium [10].

### 3.1. Does Peritoneal Endometriosis Affect Fertility?

The association between peritoneal endometriosis and infertility is controversial. The real incidence of this form of endometriosis is still unknown. According to Vercellini et al., minimal/mild endometriosis might represent a temporary phase of a process that more frequently results in the cytolysis of implanted endometrial cells [11]. The gold standard test to diagnose endometriosis is a histological examination with direct biopsy at laparoscopy. Unfortunately, peritoneal endometriosis cannot be identified by any imaging modality. The efficacy of magnetic resonance imaging (MRI) to diagnose peritoneal endometriosis is not well established [12,13]. Indeed, the diagnosis without using laparoscopy is very complex, and its prevalence among infertile women is probably underestimated. At the same time, there is no indication to perform a diagnostic laparoscopy for all women with infertility, even before In vitro fertilization (IVF) [13]. 

In healthy patients, menstrual debris is eliminated by anti-inflammatory macrophages. In the case of endometriosis, macrophages with a pro-inflammatory profile constitute the main population. The pro-inflammatory activity is allowed by a defective function of several cell types, including T helper, natural killer, and cytotoxic T cells [14].

In women with peritoneal endometriosis, an increased volume of peritoneal fluid containing more activated pro-inflammatory, chemotactic, angiogenic, and oxidative stress factors has been observed [14,15,16]. Thus, monocytes/macrophages maintain chronic inflammation, which results in adhesion formation and neoangiogenesis [17].

It seems reasonable that a hostile peritoneal and/or tubal environment may be considered a possible cause of infertility in cases of minimal-mild endometriosis. The toxic or inflammatory effects of peritoneal fluid are observed on fallopian tube function, gamete transport, embryo implantation, sperm count, and function [15]. Previous studies observed that media containing peritoneal fluid obtained from infertile women with mild endometriosis led to a decrease in the fertilization capability of oocytes and the development potential of embryos [16]. In this scenario, IVF-ICSI could act by removing oocytes from a hostile environment.

In endometriosis, an altered progesterone and estrogen signaling with a resulting progesterone resistance has been observed. This imbalance, besides increasing the severity of the inflammatory state, might decrease endometrial receptivity to embryo implantation [17]. Further endocrine and ovulatory anomalies have been observed as well, including Luteinized Unruptured Follicle (LUF) syndrome, the luteal phase defect, abnormal follicular growth, and premature luteinizing hormone peaks. Moreover, numerous studies suggest that altered implantation mechanisms may be the basis of infertility associated with peritoneal endometriosis as a consequence of a reduced expression of integrin ανβ (a cell adhesion molecule) in the implantation phase [18], reduced levels of an enzyme involved in the synthesis of a protein that coats the trophoblast on the surface of the blastocyst (L-section) [19].

In previous study analysing the impact of endometriosis on IVF cycles in women younger than 35 with minimal/ mild endometriosis, results were similar to tubal factor infertility, with the exception of fertilization rate. In the same study, in patients with the I-II stage of endometriosis, the fecundity rate dropped significantly to 4% one year after surgery. It was hypothesized that in patients surgically treated for endometriosis, the peritoneal fluid containing activated inflammatory factors might progressively increase after surgery [20].

### 3.2. Does Ovarian Endometrioma per se Affect Fertility?

To date, TVS represents the standard imaging technique for identifying ovarian endometriomas due to its high values in sensitivity (93%) and specificity (97%) when performed by an expert operator [21,22].

MRI is being used in the evaluation of patients with endometriosis as a complementary method to TVS, particularly when the clinician questions the possible presence of deep infiltrative lesions [23]. Approximately 15–44% of women suffering from endometriosis have ovarian endometriomas, and both ovaries are involved in 19–28% of cases [24,25,26,27,28]. Numerous studies evaluated the reproductive outcome after surgery for ovarian endometrioma [29,30,31,32]. Only one study evaluated the chance of spontaneous pregnancy in patients with endometriomas without a history of infertility [33]. There is growing evidence supporting the potentially detrimental effect of endometrioma per se on ovarian physiology. A systematic review (Sanchez and colleagues) suggests that the presence of an endometrioma causes ovarian damage independently from its size by mechanical stretching, compression of healthy tissue, and hampering the regular blood flow [34]. The endometriotic content might produce serious alterations to the endometriotic surrounding cells, including modifications in the expression of critical genes and genetic changes potentially inducing carcinogenesis [35,36].

The process of oxidative stress plays an important role in the impairment of the reproductive system of patients with endometrioma. Endometrioma contains high levels of cellular damage-mediating factors, proteolytic enzymes, inflammatory molecules, reactive oxygen species, and iron. The loss of balance between oxidant and antioxidant molecules in serum and in follicular fluid (FF) has been suggested to be responsible for irregular development of oocytes as a consequence of DNA and cell membrane damage associated with altered fertilisation, implantation, and embryonic development with reduced egg and embryo quality [37,38].

Ovaries with endometriomas showed an increase in fibrosis with loss of cortex-specific stroma and a lower density of follicles when compared to contralateral healthy ovaries (6.3 ± 4.1/mm^3^ vs. 25.1 ± 15.0/mm^3^) [39].

Furthermore, regular vascular networks and overall follicular maturation up to the antral stage were less frequent in the ovarian tissue surrounding the endometrioma in comparison with other ovarian cysts. Discordant data are reported about anti-müllerian hormone (AMH) levels in patients with unoperated ovarian endometrioma [40].

### 3.3. Does Deep Endometriosis Affect Fertility?

Peritoneal endometriotic lesions infiltrating to a depth of at least 5 mm beneath the peritoneal surface are defined as deep endometriosis [41]. Deep endometriotic lesions can be found in many sites throughout the pelvis, including the pelvic peritoneum, pouch of Douglas (POD), rectum, rectosigmoid, rectovaginal septum, uterosacral ligaments (USLs), vagina, bladder, and ureter. Additionally, even though rare, these lesions have also been described in extra pelvic sites. Typically, deep endometriosis is related to severe painful symptomatology such as dysmenorrhea, deep dyspareunia, chronic pelvic pain, and painful defecation. [42].

The relation between deep endometriosis and infertility is not clear, and interpretation of the available evidence is challenging. There is substantial heterogeneity in the reported sensitivity and specificity of TVS regarding the detection of deep endometriosis, irrespective of its location. In fact, diagnosis by TVS remains operator-dependent. Furthermore, authors use different terms when describing the same structures and locations.

Consequently, data on deep endometriosis are derived from studies carried out on patients undergoing surgery. In operated patients, however, only 6.5% showed deep endometriosis as the only form of the disease. The coexistence of superficial endometriotic implants, endometriomas, and pelvic adhesions was documented in 61.3%, 50.5%, and 74.2% of patients with deep endometriotic nodules, respectively [43]. Certainly, in cases where sexual intercourse is painful, the effect is due to the reduction of the coital frequency [44]. A close association between deep endometriosis and adenomyosis has been observed. In this case, the effect on fertility could be linked to the latter [45]. Limited data is available on the spontaneous fertility of women with deep endometriosis, and data on the management of deep lesions in infertile patients is scarce. Published studies show many confounding factors. Firstly, the analysis did not include patients with infertility (at least one year preoperatively attempting to get pregnant); consequently, reported data on pregnancy outcomes following surgery or ART are not reliable. The study populations are very heterogeneous and not comparable due to the different classifications adopted by various authors. Similarly, surgical techniques, mainly for colorectal endometriosis, are variable [46].

## 4. Laparoscopic Surgery

There are few studies analysing expectant management in women with endometriosis. It is estimated that with no intervention, 50% of women with mild endometriosis will conceive, 25% with moderate endometriosis, and only a few with a severe disease [3]. However, these estimates do not apply to women with infertility.

Laparoscopic surgery was classically considered the gold standard in the treatment of endometriosis-related infertility. It aims to remove visible lesions of endometriosis and restore the normal pelvic anatomy.

### 4.1. Should Minimal Mild Endometriotic Lesions Be Surgically Treated?

The rate of spontaneous pregnancy among women with endometriosis stages I–II and women with unexplained infertility is similar; thus, minimal/mild endometriosis has probably a marginal effect on fertility [46,47,48]. It is widely accepted that minimal/mild endometriosis in infertile patients may be considered equivalent to unexplained infertility and can be managed accordingly.

A meta-analysis of two randomized controlled trials (*n* = 444) showed that laparoscopic ablation or resection of minimal and mild endometriosis plus laparoscopic adhesiolysis increased ongoing pregnancy and live birth rates when compared to diagnostic laparoscopy [49]. Recently, Cochrane analysed the use of laparoscopic surgery to treat pain and infertility in patients with endometriosis. Fourteen randomized controlled trials were included. There was moderate-quality evidence that laparoscopic surgery increases the chance of pregnancies confirmed by ultrasound versus diagnostic laparoscopy only. No studies were found reporting live birth data [50].

Therefore, since diagnostic laparoscopy no longer plays a role in the diagnostic work-up of woman with infertility and that minimal mild endometriosis cannot be specifically diagnosed by imaging technique, we can conclude that the treatment of these lesions should only be done during laparoscopic surgery in infertile woman.

### 4.2. Should Ovarian Endometrioma Be Surgically Treated?

The management of endometriomas in infertile women is still debated [51,52]. If the removal of ovarian endometriomas in infertile patients is associated with a reported pregnancy rate of 37.4–50% [53,54], surgery for ovarian endometriosis is also related to damage caused by surgery. This damage can result in a worse reproductive prognosis with an increased of risk premature ovarian failure, even if the most skilful surgeons performed conservative procedures [55,56,57,58].

Size and type of surgery (stripping with coagulation, laser, suture) on ovarian endometrioma can influence the appropriateness of surgical management. As a consequence of cystectomy, ovaries showed reduced responsiveness to gonadotropin stimulation. These effects are clinically more relevant in patients with bilateral endometriomas who undergo repeated surgeries for recurrences [55,56,57,59,60,61,62,63].

In a previous study, we demonstrated that surgery for bilateral endometriomas is associated with a dramatic fall in ovarian reserve. In these patients, we observed a mean basal follicle-stimulating hormone (FSH) of 13 ± 3.5 U/L and a cancelation rate during the IVF-ICSI cycle of 28.2%. This was significantly higher when compared to patients with ovarian endometrioma(s) and no previous surgery, patients operated on for monolateral ovarian endometrioma, and women infertile for tubal factor. Surgery in bilateral ovarian endometrioma resulted in a halving of ovarian response to 4.6 ± 3.4 oocytes from 7.3 ± 4.8 in women with unoperated endometrioma and 8.8 ± 5.7 in patients with tubal factor. [64].

In all women undergoing surgery for endometriosis, the mean age at menopause was significantly lower than the mean age of menopause observed in a reference population of Italian women (45.3 ± 4.3 years versus 51.2 ± 3.8) [65]. Mainly in patients with previous surgery for bilateral endometriomas, we observed that menopause (42.1 ± 5.1 years) occurred at a relatively young age, and a high percentage of women with premature ovarian failure (36.4%) or lamenting menopausal symptoms (12.3%) [56].

## 5. Ovarian Endometrioma and Infertility: Risk of Expectant Management

There is currently insufficient data to determine whether the endometrioma-related damage to the ovarian reserve precedes or follows surgery. A study showed a possible negative effect of unoperated ovarian endometrioma during the IVF cycle related to its size. In the ovaries containing endometriomas, significantly lower numbers of follicles (>16 mm diameter) and oocytes retrieved were observed. In patients with endometriomas larger than 30 mm, endometrioma size revealed the most influential contributor to the total number of follicles and oocytes retrieved. For every millimeter of increase in endometrioma size, the predicted number of retrieved oocytes decreased by 0.667, with all other variables held constant. In the case of endometriomas < 30 mm, basal FSH concentration remains the most important prognostic factor for oocyte retrieval [66]. Subsequent studies have seen similar results for larger endometriomas (≥5 cm) at the time of IVF [67]. Therefore, both ovarian damage after surgery and the effect of unoperated endometrioma appear to be related to the size of the cysts. According to the European Society of Human Reproduction and Embryology (ESHRE) guidelines for the diagnosis and treatment of endometriosis, infertile women with endometrioma larger than 3 cm, there is no evidence that cystectomy prior to treatment with ART improves pregnancy rates [51,68,69]. In these cases, cystectomy prior to ART might be taken into consideration only to improve endometriosis-associated pain or the accessibility of follicles [13].

Women should be informed regarding the risks of reduced ovarian function after surgery for ovarian endometrioma. The decision to proceed with surgery or not should be considered carefully in patients with a previous ovarian operation, in cases of pre-existing low ovarian reserve or bilateral ovarian endometrioma.

In cases of patients with an endometrioma above 3 cm when conservative management is chosen and, therefore, to proceed with IVF, it is important to take two aspects into account: first, the cases of women who undergo IVF-ICSI with large endometriomas are not several; second, endometriomas with a diameter of more than 7–8 cm have a risk of malignancy and infection due to pick up. Recently, it has been emphasized that endometrioma diameters larger than 8 cm are associated with increasing age and long duration of disease, which can be predictive of malignant transformation and aid in identifying women at increased risk of cancer [70].

### Which Are Possible Results and Risks of Surgery for Deep Endometriosis?

Radical surgery for deep endometriosis is complex and often requires a multidisciplinary approach, including general surgeons and urologists. Even though operations are associated with pain relief, the risk of major complications and the recurrence, or persistence, rate are very high in these patients. The reported incidence of rectovaginal fistula formation, anastomotic leakage, and postoperative ureteral fistula formation ranged between 2–10%, 1–2%, and 0.5–1%, respectively [71]. When colorectal resection is performed, post-operative complications are even higher. In a series on laparoscopic treatment of deep endometriosis with segmental colorectal resection, major complications were observed in 10.4% of patients and included: anastomotic leakage (4.7%), rectovaginal fistula (2.7%), anastomotic fistula (2%), perforation (0.5%), bowel obstruction (0.5%), uroperitoneum (1.5%), ureteral fistula (1%), bladder fistula (0.5%), pelvic abscess (0.5%), sepsis (0.5%), hemoperitoneum (2%), heterologous blood transfusion (6%), and, after 30 days, urinary retention (4.7%), constipation (2.6%) and peripheral sensory disturbance (1.5%) [72]. Since the data on complications are published by groups specialising in deep endometriosis surgery, these data relate to the best of conditions and are, therefore, not generalizable.

These issues can create a conflict between the radicality of excision and the patient’s needs and desires (pregnancy and relieving painful symptoms). Therefore, a major conservative approach centered on the patient’s symptoms and needs has been advocated for in endometriosis in general and, particularly, in deep endometriosis [73].

The potential of surgery for deep endometriosis to increase the likelihood of spontaneous conception has yet to be established. Three systematic reviews of the literature reported a pregnancy rate ranging from 42 to 44% after excisional surgery for rectovaginal endometriosis [57,71,74].

However, these data must be considered with extreme caution due to the lack of a control group. Not all included patients were infertile, and both natural and IVF pregnancies were evaluated [75].

A systematic review of the literature, including only those studies which reported natural pregnancies in infertile women with deep endometriosis, showed less optimistic results (pregnancy rate 24% (95% CI 20–28%) [76].

## 6. Deep Endometriosis and Infertility: Risk of Expectant Management in Pregnancy

Decidualization of deep infiltrating endometriotic lesions, with the resultant decrease in size of a transmural endometriotic nodule, may lead to perforation, by weakening the intestinal wall, mainly during the third trimester of pregnancy. Actually, intestinal perforation [77,78,79], pneumothorax [80], and pelvic vessel rupture [81] have been reported in pregnant women with deep endometriosis. On the other hand, transmural bowel wall involvement is very rare as the intestinal mucosa usually remains intact, and intestinal perforation is a very rare complication [82].

The hormonal environment produced by pregnancy might determine significant modifications of endometriotic lesions and improve painful symptoms. In a series of three cases of patients with deep endometriosis involving recto-vaginal and recto-sigmoid tracts, achieving spontaneous pregnancies were followed up by TVS. During the third trimester, the lesions were more homogeneous with less evident limits of nodules and band-like echoes and less fibrotic-like. All patients showed complete resolution of symptoms [83].

Therefore, the chances of complications of deep endometriosis reported in pregnancy are mostly anecdotal and appear lower than complications related to the surgical treatment of the same lesions. Furthermore, pregnancy would reproduce the conditions of prolonged medical therapy and favour the regression of the lesions.

## 7. Assisted Reproductive Technology (ART)

ART for infertility associated with endometriosis was first indicated in cases of tubal factor or severe male factor infertility. Nonetheless, the general trend to delay the first childbirth, recent knowledge in ultrasound diagnosis, and progress in ART laboratory have influenced the approach to infertility-related endometriosis. Nowadays, ART represents one of the first-choice approaches and has a key role.

ART in infertile women with endometriosis posed some relevant questions. One aspect is to understand how pathology can affect the results; the second is the possible worsening of the pathology being treated and, lastly, how to integrate it with other medical/surgical treatments.

The effect of endometriosis on the success rate of ART is still not clear. Barnhart et al. conducted a meta-analysis of 22 studies from 1983 to 1997 in which they observed that overall, endometriosis significantly affects all markers of the reproductive process, resulting in a pregnancy rate that is almost halved when compared to tubal factor controls (odds ratio 0.56, 95% CI 0.44–0.70%). However, the meta-analysis did not include randomised controlled trials. The included studies were outdated and did not consider improvements in the IVF-ET performances. The outcome reported for each study varied, with some studies reporting crude pregnancy rates and others reporting clinical pregnancy and rarely live birth rates. The articles could not be categorized as to whether the endometriosis was medically or surgically treated before the initiation of IVF-ET or whether endometriotic lesions were present at the time of the IVF cycle [84].

Large databases such as Society for Assisted Reproductive Technology (SART) and Human Fertilisation and Embryology Authority (HFEA)) indicate that there is no difference in IVF-ET outcome for women with infertility-related endometriosis.

Ten years after the Bernhart meta-analysis, Harb et al. published a further meta-analysis including 27 observational studies. Interestingly, the review showed that fertilisation rates were reduced in stage I/II of endometriosis (relative risk [RR] = 0.93, 95% CI 0.87–0.99, *p* = 0.03) and a decrease in the implantation rate (RR = 0.79, 95% CI 0.67–0.93, *p* = 0.006) and clinical pregnancy rate (RR = 0.79, 95% CI 0.69–0.91, *p* = 0.0008) in women with stage III/IV endometriosis undergoing IVF treatment. A reduction of 14% in live births (RR = 0.86, 95% CI 0.68–1.08) was shown in cases of stage III/IV endometriosis. Unfortunately, there were few studies reporting live births; thus the power to detect this difference was weakened [85].

Hamdan et al. analysed the impact of ovarian endometrioma on IVF/ICSI outcomes. The authors performed a systematic review and meta-analysis on infertile women with endometrioma undergoing IVF/ICSI. Both patients who have or have not had any surgical management for endometrioma before IVF/ICSI were included. Compared with women with no endometrioma undergoing IVF/ICSI, women with endometrioma had a similar live birth rate (LBR) and clinical pregnancy rate (CPR) with a lower mean number of oocytes retrieved and a higher cycle cancellation rate compared with those without the disease (OR 2.83; 95% CI 1.32–6.06]. Unfortunately, only one study reported LBR, cancellation rate, and baseline FSH level. Furthermore, the authors did not specify whether LBR was analysed on started cycles or cycles with ET [86].

The effect on the possibility of success is linked to the severity of the pathology. The most limiting factor is the ovarian response to stimulation and, consequently, to the number of oocytes obtained. Different stages of endometriosis might interfere with IVF-ET outcomes through different mechanisms. Therefore, among the indicators to be evaluated, it would be appropriate to take into consideration the number of cycles cancelled and the number of oocytes necessary to obtain a pregnancy.

In order to evaluate the impact of endometriosis on IVF-ET, we performed a study on young women with previous surgery for endometriosis and without clinical/ultrasonographic signs of recurrence stratifying analysis based on the ASRM stage. For the moderate/severe stage, a deleterious effect on IVF-ET cycles in terms of cancellation rate, poor responsiveness, and implantation rate was observed. Stage I–II showed a significantly impaired fertilisation rate [20].

The negative association between moderate/severe endometriosis and the number of oocytes retrieved might be ascribed to the effect of previous surgical treatment [87,88]. Furthermore, pelvic adhesions can impair oocyte release from the ovary or hinder ovum pick-up or transport in these patients [89,90].

Regarding the possible worsening or recurrence of endometriosis during treatment COH for ART, the risks of worsening endometriosis might be due to higher E2 levels. On the other hand, estrogen and progesterone receptors have different expressions in eutopic and ectopic endometrium, and the E2 level is increased only for a few days [91]. Endometriosis is a chronic estrogen-dependent disease with a reported recurrence rate after conservative surgery ranging from 2% to 51% [92,93].

In order to assess whether controlled ovarian hyperstimulation (COH) for ART was associated with an increased incidence of endometriosis recurrence as documented by TVS, we analysed patients with infertility-related endometriosis (Figure 1).

Survival curves showed a cumulative recurrence rate of 28.6% in 90 patients submitted to ART after surgery for endometriosis. This was similar to the 37.9% observed in the control group (87 women never undergoing COH after surgery for endometriosis) (*p* = 0.471) [94].

Previous studies have shown that recurrence rates of endometriosis tend to be lower in women with infertility and higher in women with advanced stages of the disease or pelvic pain [60,92].

Patients with severe stages of the disease were more likely to have a recurrence in both the ART and control groups. Furthermore, patients with pelvic pain at the time of their first surgery for endometriosis showed recurrence in a shorter time span [53]. The combined approach of surgery followed by ART proved to offer higher chances of pregnancy in infertile women with endometriosis. Timing depends on the patients’ age and the severity of the disease. Thus, in infertile women with endometriosis, ART after surgery might be offered since cumulative endometriosis recurrence rates are not increased after controlled ovarian stimulation for IVF/ICSI [94,95,96].

### 7.1. Which ART Procedure Is the Most Appropriate in Patients with Endometriosis?

#### 7.1.1. Peritoneal Endometriosis

In infertile women with AFS/ASRM Stage I/II endometriosis, intrauterine insemination (IUI) with controlled ovarian stimulation is recommended as it increases live birth rates [13,97,98].

Accordingly to ESHRE guidelines (2014), after surgical treatment, clinicians may consider performing IUI with controlled ovarian stimulation within six months since pregnancy rates are similar to those achieved in unexplained infertility [13,99].

#### 7.1.2. Ovarian Endometriomas and/or Deep Infiltrating Endometriosis

IUI in moderate-to-severe endometriosis patients is not implemented according to international guidelines, as only limited data exist on treatment efficacy and safety [13]. We have no cost effectiveness studies in the use of IUI or IVF in women with severe endometriosis and/or deep endometriosis. However, given the impact of the disease on fertility status and the chances of success in women with endometriosis, in our opinion, in patients with ovarian endometrioma/s larger than 3 cm and/or deep infiltrating endometriosis ART II Level (IVF/ICSI) should be offered even in the case of tubal patency and regular seminal fluid.

IVF or ICSI might be the preferred option in these patients, in light of the greater chances of success and the opportunity to reduce the time to pregnancy. Whether this treatment strategy can be systematically offered in all patients with ovarian endometriomas and or deep endometriosis, should be investigated in a randomized controlled trial [100].

### 7.2. Which Is the Most Suitable Protocol for Ovarian Stimulation in Patients with Endometriosis, Undergoing IVF-ICSI?

We have no satisfactory data on the best stimulation protocol for women with endometriosis undergoing IVF-ICSI. However, patients with endometriosis have some peculiarities that could be relevant to ovarian stimulation. In fact, in addition to the possible reduced ovarian reserve, the pelvic cavity of women with endometriosis has a greater level of inflammation which is amplified by the stimulation and subsequent egg retrieval. Furthermore, due to the reduced quality of the oocytes, a greater number of oocytes may be required to reach the same pregnancy rate.

We must also take into account that patients often suspend hormonal therapies for pain relief to perform IVF cycles. So, we need to ensure the greatest possible efficiency of each stimulation cycle. The only indication contained in the ESHRE guidelines concerns the pre-treatment with GnRHa for three to six months [13]. This suggestion was based on a meta-analysis including only three randomized clinical trials [101]. The advantage of agonist treatment before ovarian stimulation could be due to the decidualization of minimal lesions and, therefore, the reduction of the inflammatory state of the pelvic cavity [18,57]. However, especially in patients with reduced ovarian reserve, postponing treatment for 3–6 months is not a viable option.

With a similar effect, pre-treatment with oral contraceptives might be helpful in the case of adenomyosis, but this deserves further verification [46].

The segmented cycle could be a viable option to reduce the inflammatory state. While, to increase the number of available oocytes, especially in women with low ovarian reserve, double stimulation could represent a valid choice.

The antagonist protocol allows access to GnRH-a triggering, which is key for a segmented ART approach or a duo-stim protocol. This protocol is generally better tolerated and might prevent OHSS while providing similar results [102,103].

## 8. Endometriosis Is a Chronic Pathology and Involves the Reproductive System: Fertility Preservation

Endometriosis should be taken into account as a chronic pathology. Further, it should be considered that it is prevalent in women of reproductive age, involves the reproductive system, and decreases their reproductive capacity.

Consequently, both treatment and counselling should always bear patients’ reproductive future in mind and aim at preserving the chances of pregnancy for the patient, even if not associated with infertility. This broader view on endometriosis led to a reconsideration of the role of medical therapy in the management of endometriosis and fertility preservation procedures.

Medical therapy has been shown to be effective in reducing the progression of the disease and the percentage of relapses after surgery. However, in infertile women, it did not show clear benefits and may delay more effective treatments [104].

Fertility preservation might represent a valid treatment option for women with endometriosis who are at risk of disease progression or need surgical intervention in order to increase their future reproductive chances. Options for preserving fertility in women include embryo/oocyte cryopreservation and ovarian tissue cryopreservation. Nowadays, the procedures of vitrifying both oocytes (preferable to avoid problems) and embryos is a routine clinical treatment in IVF clinics. The need for ovarian stimulation and repeated cycles, risk of infections, and abscess formation represent some disadvantages of this option. The development of pelvic abscesses after oocyte retrieval in patients with ovarian endometriomas has been reported as a rare complication [105].

Ovarian tissue cryopreservation requires a laparoscopic procedure and is easily performed during the surgical intervention for the disease. On the other hand, performing surgery to preserve fertility is more difficult and riskier. Ovarian tissue can be obtained from the cyst capsule or a site distant from the endometriotic lesion. In the case of the former, the number of eggs that can be obtained and the quality could be altered by the endometriotic pathology. In the case of the latter, the technique could produce further damage to the residual ovarian reserve [106].

The most viable path today appears to be the vitrification of oocytes. Cobo et al. published the first and largest series to date on the use of vitrified oocytes for fertility preservation in patients with endometriosis [107]. The multicentre, descriptive, and observational study included 485 women with endometriosis whose oocytes were vitrified and returned to attempt pregnancy (return rate 46.5%). The mean age at vitrification was 35.7 ± 3.7 years; 47.8% were treated after having their endometrioma surgically removed before FP. The number of vitrified oocytes per cycle (6.2 ± 5.8) was higher for the nonsurgical patients compared with the unilateral (5.0 ± 4.5) or bilateral (4.5 ± 4.4) surgery groups. The survival rate and cumulative live birth rate (CLBR) were 83.2% and 46.4%, respectively. The great difference in the CLBR among the youngest nonsurgical (~75%) versus the oldest surgical patients (~30%) suggests that young women would be the best candidates for fertility preservation before surgical treatment due to their better ovarian reserve and reproductive prognosis. In fact, as previously proved with fresh and vitrified oocytes, the more oocytes, the higher the CBLR. On the contrary, the procedure in older women would not be as effective, irrespective of the surgical treatment [107].

In a smaller retrospective study conducted on 34 women with endometrioma undergoing an oocyte cryopreservation cycle for fertility preservation before a planned ovarian cystectomy, Kim et al. observed a lower number of mature oocytes retrieved, and percentage of cryopreserved oocytes was lower in patients with bilateral endometrioma than in patients with unilateral endometrioma [108]. Despite the fact that the data did not reach statistical significance, the authors suggested that, in cases of bilateral endometrioma, fertility preservation should be carried out even when the AMH level is relatively high [108]. Moreover, this indication is also supported by the observation that the ovary site of endometrioma has greater difficulty ovulating [109]. Furthermore, it is now known that bilateral ovarian surgery will cause greater damage in terms of reduction of the ovarian reserve [64].

Regarding the number of oocytes sufficient to guarantee a future pregnancy, there is not much data. Surely the more eggs are collected, the greater the chances of pregnancy. Sunkara et al. observed that the best chance of live birth was seen with approximately 15 oocytes [110]. At the same time, more oocytes are required as the woman’s age increases [111].

There is a lack of data regarding the need for more oocytes for patients with endometriosis. As the baseline ovarian reserve is often reduced in women with endometriosis, repetitive ovarian stimulation cycles might be necessary to increase the number of available oocytes.

The effect of endometriosis on oocyte quality is controversial. Previous studies suggested a compromised quality of oocytes in women with endometriosis: women with endometriosis had oocytes with lower in-vitro maturation rates, more altered morphology, and lower cytoplasmic mitochondrial content compared with infertile women with other causes [112,113]. The fertilization rate was lower than that in the controls [84,85]. Similarly, Cobo et al. compared data obtained with vitrified oocytes obtained in endometriosis patients with a ‘historical’ control group of patients with no diagnosis of endometriosis. They observed oocyte survival, implantation, pregnancy rates, and CLBR were significantly lower than in the group without endometriosis [107,114].

On the other hand, the embryo aneuploidy rate showed similar in patients with endometriosis who underwent IVF and unaffected controls [107,114], and the fertilization rate was even higher in patients with stages III-IV [20,84]. However, recommending the preservation of fertility without distinction in all patients with endometriosis appears improper. In fact, endometriosis is a frequent pathology among women of reproductive age. The procedures necessary for preservation present a certain degree of invasiveness and are costly. Additionally, there are no indications of the adequate number of oocytes to guarantee pregnancy in patients with endometriosis.

Therefore, until we have more data in terms of cost-effectiveness, it seems appropriate to propose preservation in selected cases of patients at greater risk of ovarian compromise. In cases of bilateral ovarian endometriomas (in which the smaller one measures more than 3 cm) and patients with recurrent ovarian endometriosis, fertility preservation should be offered in any case. In cases of monolateral ovarian endometriosis, decisions should be based on both patients’ age and age-related AMH. Bilateral endometriomas with the smaller endometrioma <3 cm should be managed as monoliteral (Figure 2).

All patients diagnosed with endometriosis should have at least an annual follow-up to assess the trend of the ovarian reserve (AMH values, antral follicle count) and the size of the endometriotic cyst. This will enable it to modulate the counselling more easily; in fact, in the event of a decrease in the ovarian reserve or an increase in the cyst, the indications might change.

Given the age-related reduction in the reproductive potential in patients over 35 years of age, the indications for preserving fertility appear the same as those of other patients without endometriosis.

## 9. Propose an Algorithm

The guidelines known to date have numerous gaps, from the integration of medical/surgical treatment to the management of endometriomas larger than 3 cm and stimulation protocol to be used [13,115,116,117]. Given the lack of data on the real impact of minimal endometriosis and deep endometriosis on women’s fertility, we suggest a management algorithm for women with infertility associated with endometrioma that takes into account both the size of the endometrioma, the age of the patient, and the ovarian reserve (Figure 3). In cases of endometriomas in very symptomatic young women, refractory to medical therapy with a good ovarian reserve, laparoscopic surgery might be proposed as a first-line treatment. In all patients with a low ovarian reserve, IVF-ICSI is preferable. In order to increase the number of available oocytes, duo-stim protocols should be proposed in women with a lower prognosis.

When larger endometriomas are diagnosed in an infertile woman, medical treatment could be proposed for a variable period, more or less than three months, depending on the patient’s age, ovarian reserve, and the timing of IVF. The advantage is due to possible size reduction of ovarian cyst with better follicle visualization and decrease of inflammation. In the presence of endometriomas size > 5–6 cm, where follicle pick-up might be particularly difficult (particularly in mono ovary patients) with follicle compression, cyst aspiration could be conducted just a month before IVF-ICSI.

## 10. Conclusions

At the time of the first diagnosis of endometriosis, management must be framed throughout the patient’s life, considering the possible impact of the disease on the patient’s reproductive life. Furthermore, we must take into account the change in the reproductive habits of today’s society, such as postponing the age of the first pregnancy of the modern female population. Patients must be managed with adequate treatment and follow-up, and we must identify patients who can benefit from fertility preservation as early as possible.

Due to the impact of endometriosis on the reproductive state and the progressive nature of the pathology, in infertile women diagnosed with endometriosis, we must carry out all the strategies (medical, surgical, ART) according to the patient’s characteristics at appropriate times and ways.

Future studies on endometriosis should evaluate the cost/benefit of the fertility preservation procedure and stimulation protocols.

## Figures and Tables

**Figure 1 ijerph-19-06162-f001:**
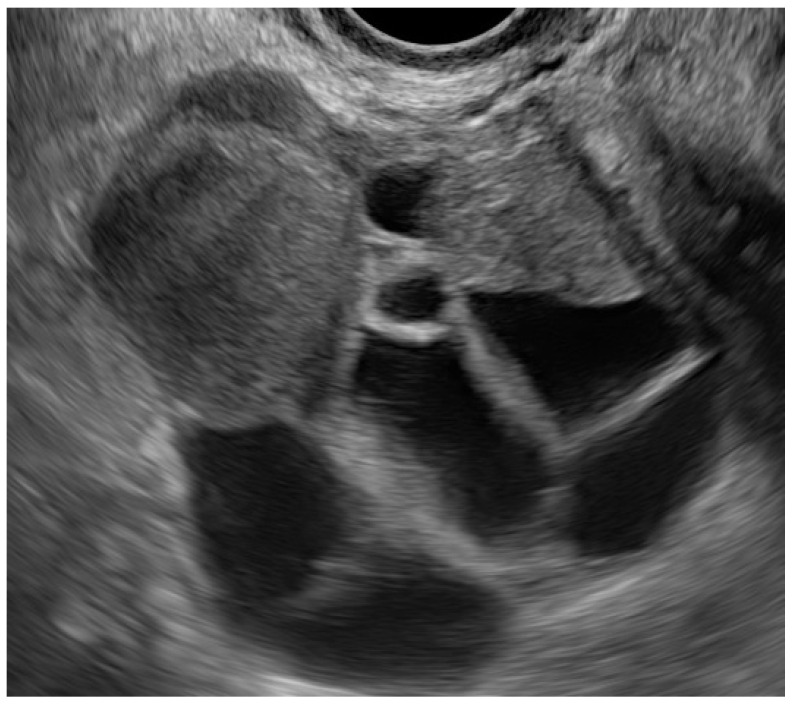
TVS ovary with ovarian endometrioma during COH.

**Figure 2 ijerph-19-06162-f002:**
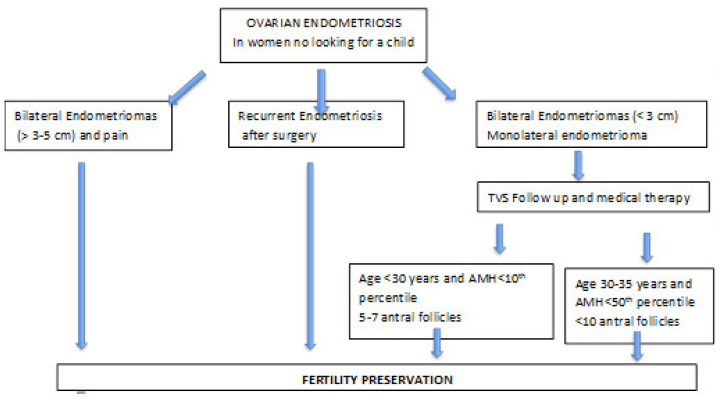
Algorithm for fertility preservation in women with endometriosis.

**Figure 3 ijerph-19-06162-f003:**
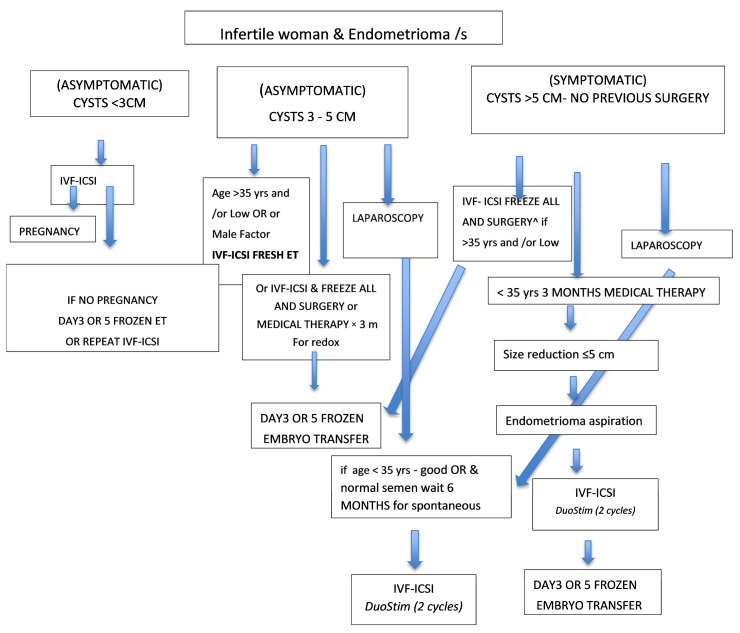
Clinical algorithms in women with ovarian endometrioma associated infertility. In the case of multiple monolateral endometriomas, the size corresponds to the sum of single diameters. In the case of bilateral endometriomas, management is based on the largest diameter. ^ in case of LOW Ovarian Reserve (OR) (AMH & AFC).

## Data Availability

Not applicable.

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
