# Peer review of "Endometriosis and Infertility: A Long-Life Approach to Preserve Reproductive Integrity"

_ijerph, 2022, doi:10.3390/ijerph19106162_

Round 1
Reviewer 1 Report
This is an interesting review article on a very important topic. The authors appear to have expertise in the clinical management of endometriosis, which increases the impact of the findings.
The text needs to be improved.
General comments:
Minor English language check would be highly recommended. The text structure should also be slightly edited, especially the too many unneeded paragraphs.
Specific examples where there is no need to change the paragraph: L34-35, L79-80, L113-114, L155-156, L231-232, L241-242, L258-259, L286-287, L298-299, L399-400, L418-419, L438-439, L474-475, L533-534, L568-569, L600-601.
You need to define all abbreviations the first time you use them (eg. FSH, CPR, RCTs, IUI needs to be explained in L410 instead of 416…). Do you really need to abbreviate things if they are not used a second time (eg. RVS in L151, ROS in L135)?
You should also be consistent with the terminology. For example, is LBR (L351) a different value from CLBR (L497)?
How about progesterone resistance? There is ample evidence supporting its role in the pathogenesis of endometriosis, while progesterone is vital to physiological endometrial decidualization and receptivity. I would suggest to also discuss the possibility of progesterone resistance being another cause of endometrial-related infertility in endometriosis patients (see these reviews for example: https://doi.org/10.3390/ijms20153822, https://doi.org/10.3390/jcm10051085, https://doi.org/10.1210/er.2014-1045).
Specific comments:
L32: The cited paper of Giudice claims a prevalence of 6-10% in the general population. Where is this 5% coming from? The authors should add a reference, especially since this number is lower than usually reported in literature.
L66: Should read: “…the evidence, although a definitive…”.
L86-88: The point is not so clear in terms of the connection between monocytes and T cells. Do you mean that T cells attract monocytes or are these 2 independent events? Please rephrase accordingly.
L125: Do you mean: “There is growing evidence supporting/demonstrating the potential detrimental effect…”?
L168: Reference is missing.
L190: Should read: “…can be managed accordingly.”.
L269: (A series on laparoscopic…): The sentence needs rephrasing.
L283: Should read: “The potential of surgery…conception has yet to be established.”.
L317-318: The sentence is not clear and needs rephrasing.
Chapter 7.2: I agree that there is no consensus regarding the best stimulation protocol for endometriosis patients. There are, however, numerous clinical trials testing several options (GnRHa, progestins, GnRHat etc). I would suggest that the authors briefly mention the available options and evidence we currently have in favor or against them, before reaching this conclusion.
L444-447: Reference is missing.
L448-449: Reference is missing.
L450 should follow L447, as they belong in the same paragraph.
L458: Should read: “Endometriosis is a chronic pathology…”.
L461-462: This sentence (which also appears in the abstract) needs rephrasing to understand the intended meaning.
L499-502: Reference is missing.
L518: I cannot find Sunkara et al. in the reference list. Please add.
L524: It is not clear what the authors are trying to say here. Oocytes are gametes. I would suggest to rephrase the sentence.
L588: Should read: ‘adequate’ and ‘follicles’.
L603: You should choose between ‘furthermore’ and ‘also’. No need for both.
Author Response
Endometriosis and infertility: a long-life approach to preserve reproductive integrity
Florence 25th April 2022
Dear Editors,
We sincerely thank you and the other two anonymous reviewers for your comments on our manuscript. Enclosed please find our replies and explanations to all of your comments and questions.
We highlighted our comments and revisions in red.
Reviewers' comments:
Reviewer: Minor English language check would be highly recommended. The text structure should also be slightly edited, especially the too many unneeded paragraphs.
Authors: The language was text was revised by a Specialized Society
Reviewer: Specific examples where there is no need to change the paragraph: L34-35, L79-80, L113-114, L155-156, L231-232, L241-242, L258-259, L286-287, L298-299, L399-400, L418-419, L438-439, L474-475, L533-534, L568-569, L600-601.
Authors: we revised the suggested lines and repetition were deleted
Reviewer: You need to define all abbreviations the first time you use them (eg. FSH, CPR, RCTs, IUI needs to be explained in L410 instead of 416…). Do you really need to abbreviate things if they are not used a second time (eg. RVS in L151, ROS in L135)?
Authors: amended
Reviewer: You should also be consistent with the terminology. For example, is LBR (L351) a different value from CLBR (L497)?
Authors: LBR is a parameter different from CLBR
Reviewer: How about progesterone resistance? There is ample evidence supporting its role in the pathogenesis of endometriosis, while progesterone is vital to physiological endometrial decidualization and receptivity. I would suggest to also discuss the possibility of progesterone resistance being another cause of endometrial-related infertility in endometriosis patients (see these reviews for example: https://doi.org/10.3390/ijms20153822, https://doi.org/10.3390/jcm10051085, https://doi.org/10.1210/er.2014-1045).
Authors: we thank you for your note. We added a short paragraph about Progesterone Resistance. ‘In endometriosis, an altered progesterone and estrogen signalling with a resulting progesterone resistance has been observed. This imbalance besides increasing the severity of the inflammatory state, might decrease endometrial receptivity to embryo implantation’
Reviewer: L32: The cited paper of Giudice claims a prevalence of 6-10% in the general population. Where is this 5% coming from? The authors should add a reference, especially since this number is lower than usually reported in literature.
Authors: amended
Reviewer:L66: Should read: “…the evidence, although a definitive…”.
Authors: amended
Reviewer:L86-88: The point is not so clear in terms of the connection between monocytes and T cells. Do you mean that T cells attract monocytes or are these 2 independent events? Please rephrase accordingly.
Authors: In healthy patients, menstrual debris are eliminated by anti-inflammatory macrophages. In cases of endometriosis, macrophages with a pro-inflammatory profile constitute the main population. The pro-inflammatory activity is allowed by a defective function of several cell types, including T helper, natural killers and cytotoxic T cells
Reviewer:L125: Do you mean: “There is growing evidence supporting/demonstrating the potential detrimental effect…”?
Authors: amended
Reviewer:L168: Reference is missing.
Authors: amended
Reviewer:L190: Should read: “…can be managed accordingly.”.
Authors: amended
Reviewer:L269: (A series on laparoscopic…): The sentence needs rephrasing.
Authors: amended
Reviewer:L283: Should read: “The potential of surgery…conception has yet to be established.”.
Authors: amended
Reviewer:L317-318: The sentence is not clear and needs rephrasing.
Authors: amended
Chapter 7.2: I agree that there is no consensus regarding the best stimulation protocol for endometriosis patients. There are, however, numerous clinical trials testing several options (GnRHa, progestins, GnRHat etc). I would suggest that the authors briefly mention the available options and evidence we currently have in favor or against them, before reaching this conclusion.
Authors: Medical treatments for endometriosis comprise GnRHa, oral contraceptives, progestins and aromatase inhibitors. These medical therapies produce contraception or subfertility, consequently are not helpful for treating endometriosis-associated infertility [Similarly, medical treatment as a neoadjuvant or adjuvant to surgical therapy has not been observed to be effective for infertility treatment [Lee D, Kim SK, Lee JR, Jee BC. Management of endometriosis-related infertility: Considerations and treatment options [published correction appears in Clin Exp Reprod Med. 2020;47(2):153]. Clin Exp Reprod Med. 2020;47(1):1-11. doi:10.5653/cerm.2019.02971].
L444-447: Reference is missing.
Authors: amended
Reviewer: L448-449: Reference is missing.
Authors: amended
Reviewer: L450 should follow L447, as they belong in the same paragraph.
Authors: amended
Reviewer: L458: Should read: “Endometriosis is a chronic pathology…”.
Authors: amended
Reviewer: L461-462: This sentence (which also appears in the abstract) needs rephrasing to understand the intended meaning.
Authors: the sentence was modified
Reviewer: L499-502: Reference is missing.
Authors: amended
Reviewer: L518: I cannot find Sunkara et al. in the reference list. Please add.
Authors: amended
Reviewer: L524: It is not clear what the authors are trying to say here. Oocytes are gametes. I would suggest to rephrase the sentence.
Authors: amended
Reviewer: L588: Should read: ‘adequate’ and ‘follicles’.
Authors: amended
L603: You should choose between ‘furthermore’ and ‘also’. No need for both.
Authors: amended
Please do not hesitate to contact us for any further details
We look forward to receiving an answer from you at your earliest convenience.
Best Regards
The authors
Reviewer 2 Report
This paper deals with a topic of great interest in scientific literature and of great interest in clinical practice. The paper is well articulated and answers the most common and controversial questions in daily practice by referring to high-level scientific literature.
I do not completely agree with the statement described in line 410 - 411 and I would like to ask the authors to discuss these reflections:
Controlled ovarian stimulation (COS) and IUI to treat infertility associated with endometriosis at any stage should be discouraged for several reasons.
Firstly, IUI as a treatment for infertility is been questioned regardless
of the indication. According to the latest NICE guideline (2017),
the procedure is not cost-beneficial for the treatment of infertility.
It can be considered only for conditions that cannot be defined as
proper infertility, such as HIV discordance, semen donation, and sexual
disturbances (Royal College of Obstetricians and Gynaecologists,
2013). Secondly, the use of IUI for endometriosis-related infertility
is not supported by scientific evidence. The procedure was initially
advocated based on low quality studies (Deaton et al., 1990; Fedele
et al., 1992; Nulsen et al., 1993; Omland et al., 1998; Tummon et al.,
1997; Werbrouck et al., 2006). To note, in some of these studies,
there was no attempt to differentiate women with unexplained
infertility and those with endometriosis stages I and II, while, in
others, endometriosis was considered together with other causes
of infertility, such as mild male factor or ovulatory dysfunctions.
Interestingly, a recent comparative non-randomized study failed to
show any difference between COS and IUI, and expectant management
(Gandhi et al., 2014). Moreover, the results of a systematic review
and meta-analysis conducted by Hughes (1997) suggest that IUI
effectiveness is halved in women with early endometriosis. Overall,
first-cycle chance of pregnancy with IVF is significantly higher than
the cumulative pregnancy rate that can be obtained after six IUI
cycles (Dmowski et al., 2002). This is somehow not surprising as
the procedure lacks a biological rationale; IVF, but not IUI, can be
expected to overcome the detrimental effects of a pelvic inflammatory
milieu.
Lastly, the risk of endometriosis recurrence appears to be increased
by IUI (Van der Houwen et al., 2014) and was reported to be higher
than after IVF D’Hooghe et al., (2006). Robust explanations for this
surprising finding are, however, lacking and further evidence is needed
(Somigliana et al., 2017)
Furthermore, I ask the authors to review or clarify the algorithm in line 563 in relation to the last box in which it is written: "and medical therapy"
Author Response
Endometriosis and infertility: a long-life approach to preserve reproductive integrity
Florence 25th April 2022
Dear Editors,
We sincerely thank you and the other two anonymous reviewers for your comments on our manuscript. Enclosed please find our replies and explanations to all of your comments and questions.
We highlighted our comments and revisions in red.
Reviewer 2: I do not completely agree with the statement described in line 410 - 411 and I would like to ask the authors to discuss these reflections:
Authors: We agree with you about this issue. IUI add very success in infertile patients in general and the benefits are even reduced with endometriosis. We modified the sentence to narrow down the scope of recommendations
--- Given the lack of data on the real impact of minimal endometriosis and deep endome-triosis on women's fertility, we suggest a management algorithm for women with in-fertility associated with endometrioma that takes into account both the size of the endometrioma, the age of the patient and the ovarian reserve
Furthermore, I ask the authors to review or clarify the algorithm in line 563 in relation to the last box in which it is written: "and medical therapy"
Authors: the algorithm was modified
Please do not hesitate to contact us for any further details
We look forward to receiving an answer from you at your earliest convenience.
Best Regards
The authors
Round 2
Reviewer 1 Report
The authors' modifications have greatly improved the quality and comprehensiveness of the text.
This is an interesting and well done review article and I would recommend it for publication by the IJERPH.